# The short- and long-term temporal relation between falls and concern about falling in older adults without a recent history of falling

**Roel H. A. Weijer**[1], **Marco J. M. Hoozemans**[1], **Onno G. Meijer**[1,2], **Jaap H. van Dieën**[1], **Mirjam Pijnappels**[1]*

**1** Department of Human Movement Sciences, Amsterdam Movement Sciences, Vrije Universiteit Amsterdam, Amsterdam, The Netherlands, **2** Orthopaedic Biomechanics Laboratory, Fujian Medical University, Quanzhou, Fujian, P.R. China

* m.pijnappels@vu.nl

**Data Availability Statement:** The data file containing data on which this manuscript has been based will be made available from the DANS

## Abstract

### Background and aim

The reciprocal relation between falling and concern about falling is complex and not well understood. We aimed to determine whether concern about falling increases after a fall and whether concern about falling increases the odds of future falls in community-dwelling older adults without a recent fall history.

### Methods

We selected 118 community-dwelling older adults (mean age: 71.4 (SD: 5.3) years) without a self-reported history of falling, one year prior to baseline assessment, from the one-year VIBE cohort for analyses. On a monthly basis, we recorded concern about falling (using the Falls Efficacy Scale-International, FES-I), as well as the occurrence of falls (through questionnaires and telephone calls). We determined 1) whether falling predicts an increase in concern about falling and 2) whether a high concern about falling is predictive of falling. Standard linear (fixed-effects) regression and mixed effects regression analyses were performed over long-term, i.e. one year, and short-term, i.e. one-month, intervals, respectively and were adjusted for gender, age and physical activity (quantified as the average total walking duration per day). Analyses were performed separately for all reported falls and for injurious falls only.

### Results

High concern about falling at baseline did not predict falls over the course of one year, nor over the course of one month. Furthermore, falls in between baseline assessment and one year thereafter did not predict increased concern about falling from baseline to one year later, independent of whether all falls or only injurious falls were considered. However, falls, either all or injurious only, happening somewhere over the course of a one-month interval, significantly predicted small increases in concern about falling (1.49 FES-I points, 95% CI

repository: https://doi.org/10.17026%2Fdans-xct-9tu2.

**Funding:** This research was funded by a VIDI grant (no. 91714344) from the Dutch Organization for Scientific Research (NWO), www.nwo.nl. The grant was rewarded to M. Pijnappels. The funder had no role in study design, data collection and analysis, decision to publish, or preparation of the manuscript.

**Competing interests:** The authors have declared that no competing interests exist.

[0.74, 2.25], p<0.001 for all falls; 2.60 FES-I points, 95% CI [1.55, 3.64], p<0.001 for injurious falls) from the start to the end of that one-month interval.

## Conclusion

Older adults without a recent history of falling seem to be resilient against developing concern about falling after having fallen, resulting in a short-term temporary effect of falling on concern about falling. Furthermore, we found no evidence that a high concern about falling predicts future falls over a one-month or a one-year follow-up period, suggesting that concern is not a primary cause for falls in older adults without a recent history of falling.

## Introduction

The relation between falls and concern about falling is not well understood. This may be so because concern about falling is ill-defined and in practice assessed using different questionnaires that actually measure a range of psychological constructs, including for instance fear [1]. Also, different populations have been the focus of investigation, ranging from community dwelling older adults without a recent history of falling, to people who may have entered a commonly suggested vicious cycle in which a fall increases concern about falling, which in turn promotes falling [2, 3]. So far, however, it has remained unclear if a fall increases concern and/or if high concern increases fall incidence. This is the topic of the present study, which focuses on concern about falling, as measured with the Falls Efficacy Scale International (FES-I), a questionnaire that has frequently been used in fall-related research and has been translated into many different languages [4]. Moreover, we studied community dwelling elderly without a recent history of falling. A better understanding of the relation between falls and concern about falling is not only theoretically relevant but may help to optimize fall prevention in community-dwelling older adults.

Pathways through which concern about falling may affect the risk of falling have been described previously. Attention allocation while walking may be altered in response to concern about falling [5] and hence threatening objects in the surrounding receive more visual and cognitive attention [6]. This adaptation seems to give more time to anticipate obstacles, but becomes counterproductive in more complex environments with multiple obstacles, especially while dual-tasking [7, 8], and may lead to trips and falls [9, 10]. Furthermore, concern about falling can indirectly lead to falling through activity restriction and thereby induced physical decline [11, 12]. It should be noted that it may take time for physical decline to become so severe that it increases fall risk.

Delbaere and colleagues [13, 14] recorded falls every month and concern about falling, measured with the FES-I, every three months in a group of older adults (70–90 years of age), some of whom had a history of falling. They showed that concern about falling increased over time in these older adults. This was not different for groups of participants who did not fall, fell once or fell multiple times, nor for people who experienced injurious falls [14]. The latter may be surprising as it may be expected that falls with serious negative health effects are more likely to increase concern about falling than relatively harmless falls. In the same population in which Delbaere and colleagues [14] showed that falls did not affect concern about falling, they showed that concern about falling was predictive of falls in the following year [13].

To our knowledge, the studies by Delbaere and colleagues [13, 14] are the only studies that investigated both directions of the relation between falls and concern about falling, measured

with the FES-I, in the same population. However, they evaluated each direction of the association separately and with different statistical analyses, hampering a direct comparison of the interrelations between falls and concern about falling. Moreover, a large part of their participants had already fallen before the study started. This makes it difficult to accurately assess the direction of the relation between falls and concern about falling. Concern about falling in people who have recently fallen may be differently affected by a subsequent fall and people who have recently fallen may be prone to fall in the future due to other factors underlying both falls and concern about falling, which would bias any findings on the relationship between concern about falling and future falls. Therefore, in our study, we only included participants who had not fallen in the previous year.

The aim of this paper was to determine whether falling negatively affects concern about falling and whether concern about falling is predictive of experiencing a future fall in older adults without a recent history of falling. We first evaluated these relations over a long-term period, i.e. one year, as this timeframe is most often used in fall prediction models and will help to place our results in context of findings of other studies. Next, since a one-year interval is rather long and the effect of a fall on concern about falling may recover or become more pronounced within this time period, we also evaluated the short-term relation over (subsequent) one-month intervals. We performed all analyses with all types of falls and with only injurious falls, as we believe that injurious falls may have a more pronounced effect on concern about falling than less harmful falls. We hypothesized that falls will increase older people's concern about falling when performing daily activities and that this increase in concern is strongest when assessed over a one-month interval. We also hypothesized that a high concern about falling is predictive of future falls over both one-month and one-year intervals.

## Methods

We analyzed data of 118 out of 287 community dwelling older adults who participated in the "Veilig in Beweging blijven" (VIBE) study, which translates to "Safely remaining active" and was ongoing from 2017 to 2018. Participants were community-dwelling older adults, who were recruited by flyers and newsletter advertisements in the Netherlands in 2017. They were included in the study if they were 65 years of age or older, if their Mini Mental State Exam (MMSE) [15] score exceeded 19 out of 30 points and if they were able to walk at least 20 m, with walking aid if needed, without becoming short of breath or suffering chest pain. The ethical committee of the Faculty of Behavioural and Movement Sciences of the Vrije Universiteit Amsterdam approved the protocol (VCWE-2016-129) and all participants signed an informed consent form.

### Fall history

At baseline, participants were asked how often they had fallen in the previous year (Fig 1). A fall was defined as an unintentional change in position resulting in coming to rest at a lower level or on the ground [16]. Participants who reported one or more retrospective falls were excluded from the present analysis, leading to the inclusion of 118 out of 287 participants.

### Fall incidence

Falls (see definition of a fall under *Fall history)* were recorded during the long-term study period of one year after baseline assessment (Fig 1). Participants were instructed to record any fall they experienced in a fall diary and, additionally, they were called each month by telephone to ask them if they had fallen. When participants indicated that they had fallen, they were asked to indicate the date of the fall and any related injury.

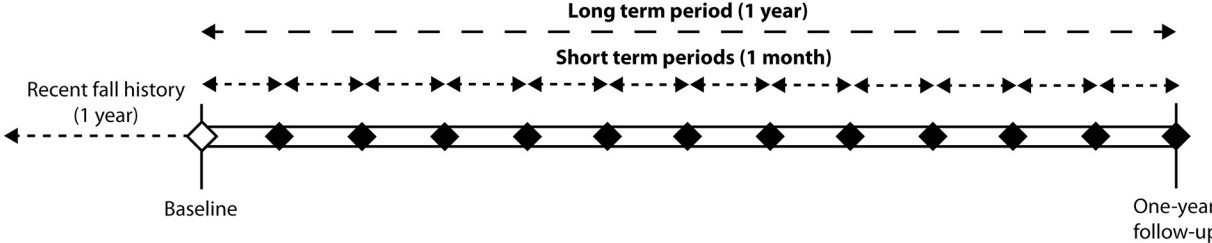

**Fig 1. Timeline of study assessments.** Short-term and long-term periods are defined to determine whether concern about falling at the start of such a period is predictive for falling within such a period and whether falling within such a period is predictive of having increased concern about falling at the end of that period compared to the start of that period. At baseline, a participant's first assessment, participants reported any falls they had experienced in the previous year and physical activity during one-week, age and gender were recorded.

## Concern about falling

All participants received a set of questionnaires, either on paper or electronically, each month over a one-year period for a total of 13 questionnaires (Fig 1). They were instructed to fill out the questionnaires independently at home. The set of questionnaires included the FES-I [4], which is a self-report questionnaire measuring concern about falling during everyday activities. It contains 16 items (e.g., "*How concerned are you that you might fall if you are cleaning the house*"), which can be answered on a scale from 1, not at all concerned, to 4, very concerned. The combined score of these items can range between 16, not at all concerned, and 64, very concerned. The Dutch version was shown to be valid and reliable with an excellent internal consistency ($\alpha = 0.96$) and four-weeks test-retest reliability (intra-class correlation coefficient = 0.82) in a population that included fallers and non-fallers [17]. The final three questions of the FES-I were not recorded in 25% of the questionnaires because they were accidentally missing in the electronic version of the questionnaire that was send during the first period of the current study. A maximum of four missing items were replaced with the average of the completed items [17].

## Covariates

Covariates included age, gender, and the average walking duration in minutes per day as measured with a trunk-worn inertial sensor. Age and gender are known to be associated with both fall frequency and concern about falling [4, 18], hence they may bias the results of our analyses if not accounted for.

The amount of walking activity may also affect our findings, as more active people may be more exposed to situations in which they are at risk of falling, but more active people may also have a better estimate of their physical ability [19] and this estimate may affect the level of concern a person has about falling. We therefore included walking duration per day to control for exposure to balance threats and to control for experience through which concern about falling may be updated. Walking duration per day was determined from inertial sensor data collected directly following the baseline and one-year follow-up assessments for seven times 24 consecutive hours. The tri-axial inertial sensor (DynaPort MoveMonitorPlus, McRoberts, The Hague, The Netherlands) assessed accelerations in vertical, horizontal and mediolateral directions, with a sample rate of 100 Hz and had a range of -8g to +8g. The participants were instructed to

wear the inertial sensors at all times, except during aquatic activities (e.g., taking a shower). The inertial sensor was placed dorsally on the trunk at the level of L3 using an elastic band. Locomotion bouts were detected using activity classification algorithms developed by the manufacturer [20]. The duration of these walking bouts was determined from days on which the monitor was worn for at least 12 hours, which were considered to be valid days. Participants needed at least two valid days for their walking duration to be validly determined, otherwise we considered it to be missing [21, 22]. The average walking duration per day was defined as the mean of the total duration of walking on valid days divided by the total recording time, independent of whether the device was worn or not, on valid days. As the battery of the devices was sometimes depleted several minutes before the end of the assessment, this method ensured that all walking duration data were comparable between participants.

Age and walking duration per day were z-transformed.

## Population descriptives

The Quickscreen fall risk assessment tool was used to indicate fall risk at the start of the study [23]. The Quickscreen consists of the assessment of physical performance, fall history and use of medication. Based on the number of identified risk factors, participants were divided into one of four subgroups: very- low (0 or 1 risk factor), low (2 or 3 risk factors), moderate (4 risk factors) or high (5 or more risk factors) fall risk, as described in Tiedemann and colleagues 2010.

Participants' handgrip strength was assessed with a dynamometer [24]. They were instructed to squeeze the dynamometer as forceful as possible, twice with each hand, while standing and holding their arms stretched downwards by their side without touching their body. We calculated the mean peak force (in kilograms) from the hands and summed the highest scores of each hand.

Number of comorbidities, the use of walking aids and education were self-reported by participants."

## Statistical analysis

Statistical analyses were aimed at determining whether concern about falling at the start of a short-term (one month) or the long-term (12-month) period is predictive for falling within such a period and whether falling within such a period is predictive of having an increase in concern about falling at the end of that period compared to the start of that period. All analyses were performed twice: first with all reported falls and then with injurious falls only. For the short-term (one-month), interval, we only analyzed data up to and including the assessment at the end of a one-month period during which a person had fallen for the first time. Any subsequent assessments were excluded from the analysis as that person would no longer be a person without a recent history of falling at that time. In all models, we considered accounting for the covariates age, gender and average total walking duration as determined from one-week inertial sensor monitoring. However, these covariates were only added to the models if their correlation with the predictor variable was less or equal to r = 0.6 (to prevent collinearity) and if including the covariate in the model changed the regression coefficient of the main predictor variable (either falling or concern about falling) by 10 percent or more. "Crude" models, which were not adjusted for confounding by covariates, are presented in the S2 Table.

**Is concern about falling increased after a fall?.** First, we assessed whether concern about falling was increased after a fall, when comparing the concern about falling at the start of either the short-term (one month) period or the long-term (one year) period with the end of that period. For the long-term interval, we fitted a linear (fixed-effects) regression model (see Eq 1)

that predicted changes in concern about falling that occurred between baseline and one year later, depending on whether people had fallen at least once in between these measurements. FES-I at baseline was a predictor for FES-I at the end of the study (one year after baseline) such that the regression coefficient of falling would indicate a change in FES-I score from start to the end of the whole one year study.

$$FESI_{Fi} = \beta_0 + \beta_1 Fall_i + \beta_2 FESI_{Bi} + \beta_n Cov_{n,i} + \varepsilon_i \tag{1}$$

where $FESI_{Fi}$ is the FES-I score at the end of the study (one year after baseline) for participant $i$, $Fall_i$ is a dichotomous variable which equals 0 if participant $i$ had not fallen and 1 if he or she had fallen, $FESI_{Bi}$ is the FES-I score at the baseline measurement for participant $i$, $\beta_{0\ldots n}$ are regression coefficients estimated with maximum likelihood, $Cov_{ni}$ are potential covariates 1 through $n$ for participant $i$ and $\varepsilon_i$ is the residual error for participant $i$.

For the short-term interval, we used a mixed effects linear regression model [25] (see Eq 2), to test whether concern about falling increased from one month to the next when a person had fallen in between assessments.

$$FESI_{i,j+1} = \beta_{0,i} + \beta_1 Fall_{i,j} + \beta_2 FESI_{i,j} + \beta_n Cov_{n,i,j} + \varepsilon_{i,j} \tag{2}$$

where $FESI_{i,j+1}$ is the FES-I score for participant $i$ at the start of month $j+1$, $Fall_{i,j}$ is a dichotomous variable which equals 0 if participant $i$ did not fall and 1 if he or she did fall in between the start of month $j$ and the start of month $j+1$, $FESI_{i,j}$ is the FES-I score for participant $i$ at the start of month $j$, $\beta_{1\ldots n}$ are regression coefficients estimated with maximum likelihood, $\beta_{0,i}$ is a regression intercept estimated for participant $i$ with maximum likelihood, i.e. random intercept, $Cov_{n,i,j}$ are potential covariates 1 through $n$ for participant $i$ at the start of month $j$ and $\varepsilon_{ii}$ is the residual error for participant $i$ at the start of month $j$.

**Does a high concern about falling increase the odds of falling?.**   Next, we assessed whether the odds of falling was increased with the level of concern about falling, over a long-term, i.e. one-year, and short-term, i.e. one-month, interval. For the long-term interval, we fitted a (fixed-effects) logistic regression model (see Eq 3) that predicted the probability of falling during a one-year follow-up depending on participants' concern about falling at the start of the follow-up period.

$$p(Fall)_i = \frac{1}{\sqrt{1 + e^{-(\beta_0 + \beta_1 FESI_{Bi} + \beta_n Cov_{n,i} + \varepsilon_i)}}} \tag{3}$$

where $p(Fall)_i$ is the probability of falling for participant $i$ in the one-year follow-up and all other elements correspond to the elements described in Eq 1.

For the short-term interval, we used a mixed effects logistic regression model [26] (see Eq 4), to test whether concern about falling increased over one month when a person had fallen in between assessments.

$$p(Fall)_{i,j} = \frac{1}{\sqrt{1 + e^{-(\beta_{0,i} + \beta_1 FESI_{i,j} + \beta_n Cov_{n,i,j} + \varepsilon_{i,j})}}} \tag{4}$$

where $p(Fall)_{i,j}$ is the probability of falling for participant $i$ in between the start of month $j$ and the start of month $j+1$ and all other elements correspond to the elements described in Eq 2.

All analyses were performed in R (R Core Team (2014), Vienna, Austria). Alpha was set at 0.05 and 95% Confidence Intervals (CI) were determined. Regression coefficients ($\beta$) from logistic models were converted to Odds Ratios (OR = $e^\beta$).

## Results

Characteristics of the 118 participants that were selected for analyses are presented in Table 1. The group with a history of falling, whom we excluded from our analyses (n = 114), was of similar age (71.7 (5.76)), had a similar proportion of women (79 (69.3%) women), but was more concerned about falling (median FES-I: 20 [18–23]) and fell more often during the one-

**Table 1. Participant characteristics.**

|  | N = 118 | |
| --- | --- | --- |
| Age, years | 71.4 (5.3) | |
| Female, n (%) | 82 (69.5) | |
| Body height, cm | 169.0 (8.3) | |
| Body weight, kg | 73.8 (12.6) | |
| Baseline Quickscreen, n (%)[1] | | |
| Very low fall risk | 45 (38.1) | |
| Low fall risk | 60 (50.8) | |
| Moderate fall risk | 4 (3.4) | |
| High fall risk | 3 (2.5) | |
| Combined hand grip strength, kg | 60.4 (16.7) | |
| Baseline daily walking duration, min/day | 84.6 (31.4) | |
| One-year change in daily walking duration, min/day | -0.2 (19.9) | |
| Use of walking aid, n (%) | 3 (2.5)[2] | |
| Highest achieved education, n (%) | | |
| Higher education | 96 (81.4) | |
| Lower secondary education | 19 (16.1) | |
| Primary education | 3 (2.5) | |
| MMSE score, median [IQR] | 28 [28, 29] | |
| Self-reported comorbidities, n (%) | | |
| Diabetes | 5 (4.2) | |
| High blood pressure | 35 (29.7) | |
| Low blood pressure | 6 (5.1) | |
| Cerebral infarction | 5 (4.2) | |
| Myocardial infarction | 9 (7.6) | |
| Thyroid condition | 6 (5.1) | |
| Asthma or Chronic Obstructive Pulmonary Disease | 9 (7.6) | |
| Pain in joints | 50 (42.4) | |
| Osteoporosis | 15 (12.7)[1] | |
| Time between completion of monthly follow-up questionnaires, median [IQR], days | 30 [28, 35] | |
| Baseline FES-I score, median [IQR] | 18 [17, 21] | |
| One-year change in FES-I score, median [IQR] | 0 [−1, 1] | |
| Number of falls experienced during the one-year study period, n (%) | All falls | Injurious falls |
| None | 58 (49.2) | 78 (66.1) |
| One | 30 (25.4) | 31 (26.3) |
| Two or more | 30 (25.4) | 9 (7.6) |

All values are means (standard deviation) assessed at baseline unless otherwise noted. IQR = Inter quartile range.

Changes in walking duration and FES-I scores from baseline assessment to one-year later are presented, positive values represent increases, negative values represent decreases.

[1] information on six participants was missing.

[2] information on one participant was missing.

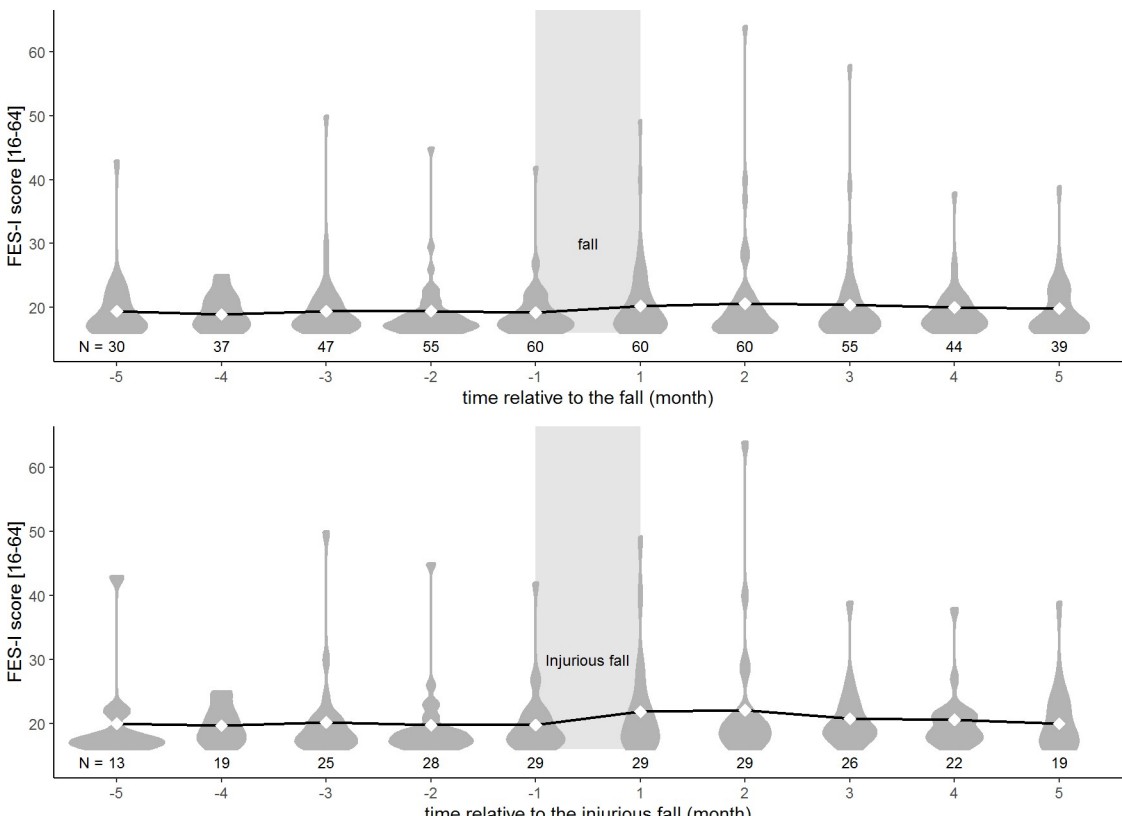

**Fig 2. Progression of concern about falling before and after a fall.** Distribution of FES-I scores per month are displayed in dark grey relative to the months in between which the participant experienced his or her first fall (area indicated in light gray) (-1 = last monthly questionnaire before the fall, +1 = first monthly questionnaire after the fall). Diamonds and solid line represent the mean FES-I scores and the progression of these mean FES-I scores, respectively. The dark grey areas represent the distribution of FES-I scores per month. The upper figure displays FES-I progression before and after any type of fall. The lower figure displays FES-I progression before and after an injurious fall. FES-I scores of people who fell more than once are only included until the first monthly questionnaire after the first fall. Sample size (N) is indicated for each month.

year study period (number of fallers: 67 (58.8%), number of people with injurious falls: 41 (36.0%)) than the people without a recent history of falling (n = 118, average age: 71.4 (5.3), woman: 82 (69.5%), median FES-I: 18 [17–21], number of people with falls: 60 (49.2%), number of people with injurious falls: 40 (33.9%)). The development of concern about falling five monthly questionnaires before and five monthly questionnaires after either a non-injurious fall or a injurious fall is displayed in Fig 2. From this figure it can be seen that in the two months following a fall the distribution of FES-I scores after the fall was wider than before a fall, with a longer tail towards high (more concerned) scores.

### Is concern about falling increased after a fall?

Over a long-term, i.e. one year, interval, falling (whether or not with an injury) did not significantly change people's concern about falling from baseline to one-year after baseline ($\beta_{\text{fall}}$ = -0.01, 95% CI [-0.88, 0.85], points on the FES-I, p = 0.974; $\beta_{\text{injuriousfall}}$ = -0.15, 95% CI[-1.06, 0.76], points on the FES-I, p = 0.747). Over a short-term interval, we found that after any type of fall, concern about falling significantly increased ($\beta_{\text{Fall}}$ = 1.49, 95% CI [0.74, 2.25] points on the FES-I, p < 0.001) and a larger significant increase was observed after an injurious fall ($\beta_{\text{injuriousfall}}$ = 2.60, 95% CI [1.55, 3.64], points on the FES-I, p < 0.001).

### Does a high concern about falling increase the odds of falling?

Concern about falling at baseline did not significantly increase or decrease the odds of experiencing a fall during the year following the baseline measurement, neither for any type of fall (Odds Ratio (OR) = 1.08, 95% CI [0.97, 1.22], p = 0.173), nor for an injurious fall (OR = 1.07, 95% CI [0.97, 1.20], p = 0.177). Similarly, over a short-term, i.e. one month, follow-up interval, concern about falling did not significantly increase or decrease the odds of falling in the next month, neither for any type of fall (OR = 1.02, 95% CI [0.95, 1.09], p = 0.501) nor for an injurious fall (OR = 1.07, 95% CI [0.96, 1.17], p = 0.171).

A summary of the results can be found in Table 2.

## Discussion

The aim of this study was to determine whether falling negatively affects concern about falling and whether concern about falling is predictive of experiencing a future fall in older adults without a recent history of falling. We found that people were more concerned about falling after a fall, injurious or not, but that this increase in concern was only observable over periods of up to a month and not over periods of up to a year. We found no evidence that concern about falling predicts falls, neither over a one-month interval nor over a one-year interval.

Our findings on the relations between falls and concern about falling in both directions appear to be in contrast with the studies by Delbaere and colleagues [13, 14]. They showed that concern about falling did not develop differently in groups of people who had not fallen or had fallen (multiple times). Possibly, the short-term effects of a fall on concern about falling that we observed in one-month intervals was washed out within the three months intervals of Delbaere and colleagues [14]. In the opposite direction of the relation, they showed that concern about falling was predictive of future falls [13] which is in contrast to our findings. These discrepancies between our study and the studies by Delbaere and colleagues [13, 14] may be explained by the fact that we excluded people with a self-reported history of falling in the previous year, which we deem a strength of our study and necessary to accurately assess the relationship between falling and concern about falling.

First, the selection of participants without a recent history of falling is reflected in our FES-I scores, which indicate that the participants of Delbaere and colleagues, including people with a recent fall history, were more concerned (mean FES-I 22.6 with SD 6.4 [14]) than our participants (mean FES-I 19.2 with SD 3.7). Perhaps, a proportion of their participants was already concerned about falling at baseline and did not become more concerned after a subsequent fall. Likewise, our population may have been too unconcerned about falling, and the range of FES-I scores too limited, for us to replicate their results.

**Table 2. Relation between falling and concern about falling over long-term and short-term intervals.**

| | All falls beta/OR [95% CI] | p-value | Injurious falls beta/OR [95% CI] | p-value |
|---|---|---|---|---|
| Fall → ΔFES-I (beta) | | | | |
| Long-term (1 year) | -0.01 [-0.88, 0.85][1,2,3] | 0.974 | -0.15 [-1.06, 0.76][1,3] | 0.747 |
| Short-term (1 month) | 1.49 [0.74, 2.25] | <0.001* | 2.60 [1.55, 3.64] | <0.001* |
| FES-I → Fall (OR) | | | | |
| Long-term (1 year) | 1.08 [0.97, 1.22] | 0.173 | 1.07 [0.97, 1.20] | 0.177 |
| Short-term (1 month) | 1.02 [0.95, 1.09] | 0.501 | 1.07 [0.96, 1.17][1] | 0.171 |

This table shows the regression coefficients (Beta, or change in FES-I score) or Odds Ratio (OR) of the long term, determined with linear fixed effect regression models, and the short term, determined with linear mixed models, effects of the main independent variables (before the arrows) on the dependent variables (after the arrow). The table includes results from analyses with all falls and only Injurious falls.

* indicates statistical significance at alpha = 0.050; The effects are adjusted for age [1], gender [2] and walking duration per day [3] as described in the methods section.

Second, our selected population consists mostly of participants with good physiological capacity as indicated by the high percentages of participants in the very low and low fall risk groups determined from the Quickscreen fall risk assessment. As suggested by Delbaere and colleagues [13], the relationship between falling and concern about falling may be dependent on physiological fall risk, because concern about falling may increase the risk of trips and thereby falls. Hence, people with good physiological capacity may be better able to compensate for this increased risk than people with poor physiological capacity. This may be a reason why we did not find that concern about falling was predictive of falling in our population. We do believe that our study population was a good representation of community dwelling older adults without a recent history of falling as these people are unlikely to have high concern about falling.

The question remains whether concern about falling can be considered as a primary cause for falls or whether it differs between people with or without a recent history of falling. Therefore, we performed the same analyses described in this manuscript on the participants with a recent history of falling (see S2 Table for population descriptives) and found no significant associations between falls and concern about falling, neither in either direction nor for the type of falls, injurious or all falls (S3 Table). Hence, participants without a recent history of falling had increased concern about falling over one month when they had fallen in between the start and end of that period (Table 2), but people with a recent history of falling did not (S2 Table). This may indicate that concern in participants with a recent history of falling was already elevated and was not further affected after a subsequent fall, conform the findings by Delbaere and colleagues [14]. These comparisons further support our belief that self-reported history of falling may either bias findings concerning the relationship between falls and concern about falling, and that this relationship is different for people with or without a recent history of falling.

Moreover, to evaluate the association between recent fall history and concern about falling, we performed linear regression analyses similarly to the long-term, i.e., one-year, models described in the method section. In one model recent fall history was associated with concern about falling at baseline and we compared its results with a second model in which having fallen determined from the one-year monthly follow-up predicted concern about falling at the end of that follow-up (not adjusted for concern about falling at baseline). The results and comparison of these results indicated that history of falling was associated with high concern about falling at baseline ($\beta_{fall}$ = 1.79, 95% CI [0.64, 2.93], points on the FES-I, p = 0.002), whereas having fallen during the one-year follow-up was not associated with concern about falling one year after baseline ($\beta_{fall}$ = 1.05, 95% CI [-0.28, 2.38], points on the FES-I, p = 0.123) and ($\beta_{injuriousfall}$ = 0.86, 95% CI [-0.53, 2.25], points on the FES-I, p = 0.225). Recall bias may be an explanation for this finding, as recall bias is likely to be more pronounced when the recall period is large, i.e., when asking about recent fall history retrospectively instead of assessing falls prospectively using a monthly follow-up. Concern about falling may possibly lead to recall bias, such that people with high concern about falling recall more falls than people with low concern about falling. This recall bias stresses the need for prospective fall assessments using monthly telephone contact or fall diaries, such as we employed in our study, to validly record falls in older adults in future studies.

Although Hadjistavropoulos and colleagues [12] argue that fear of falling is different from falls related self-efficacy and concern about falling, measured with the FES-I, these constructs are often used interchangeably in studies on fall-related psychological concern. Therefore, it may be relevant to compare our results to findings in literature that assessed fear of falling or confidence. Friedman and colleagues [2] performed a study in which they investigated and compared both directions of the relationship between falls and fear of falling over time. They found that people developed fear of falling after a fall, while using a 20-months follow-up

interval, and that fear of falling predicted future falls. Their findings seem to be in contrast to our finding as we did not observe an increase in concern about falling over a one-year period when participants fell in between assessments, nor did we find that concern about falling was predictive of future falls. However, in their study they recorded falls by asking people whether they had fallen in the past 12 months at 20-month follow-up. This method of recording falls is more likely to be biased by someone's fear or concern of falling than when falls are monitored every month, potentially resulting in differential misclassification.

In a longitudinal study in a population comparable to ours, with an exception of fall history, Hajistavropoulos and colleagues [27] found that confidence to perform daily-life activities without falling, measured at baseline, predicted falls in the following 6 months. On average their participants scored a nine out of ten on the Falls Efficacy Scale (FES), indicating that they were very confident in carrying out daily-life activities without falling [28]. Hence, they are comparable to our participants who scored on average a 19 on the FES-I scale from 16, not concerned, to 64, very concerned. The main difference between our study and their study is that they used the FES, which measures confidence, while we used the FES-I, which measures concern. As mentioned earlier, Hadjistavropoulos and colleagues [12] argue that the concepts of concern and confidence are not the same. Perhaps the difference between the concepts of concern and confidence is the reason why our results do not support the findings of Hajistav-ropoulos and colleagues [27].

Zijlstra and colleagues [29] showed that an intervention that reduced concern about falling also reduced fall prevalence in community dwelling older adults. Their findings could be interpreted to suggest that concern about falling is associated with future falls. However, their intervention also intervened on other potential determinants of fall risk. This may explain the apparent disparity with our findings.

Despite the strength of our study to exclude people with a recent history of falling it is possible that a fall before the year prior to the baseline assessment has had a lasting influence on concern about falling. Such an effect may have influenced our results as we did not assess fall history for more than one year before baseline assessment. However, since we found no evidence of long-lasting effects of falls on concern about falling, it is unlikely that fall history from the period of earlier than one year before the start of our study has influenced our findings substantially.

A limitation of our study is that we asked participants to fill out the FES-I questionnaire each month, for a total of 13 questionnaires over 12 months, to study changes in concern about falling shortly following a fall. A possible disadvantage of this approach is that participants may have become bored with filling out the questionnaires. This may have rendered the questionnaires less sensitive to small changes, as people would simply fill out the questionnaire similarly each month, unless something had happened that strongly affected their level of concern about falling. Hence, a slow gradual change in concern about falling, independent of falls, would not be detected. This could mask the short-term effect of concern about falling on falls, as it relies on an accurate assessment of concern about falling, which according to Delbaere and colleagues increases gradually over time [14]. However, although we indeed did not observe any effect of the level of concern about falling on the short-term odds of falling, we also did not observe an effect of concern about falling on the long-term odds of falling.

Another limitation is that, in 25% of the questionnaires used in this study, three items were accidentally missing and replaced by the mean of the remaining items if no more than four items were missing in total. The missing items were "Walking on an uneven surface", "Walking up or down a slope" and "Going out to a social event". To investigate if these missing items influenced the total score of the questionnaires, we compared complete questionnaires and questionnaires with missing items. We did this only for participants who did not experience

any falls during the follow-up and who filled in both questionnaires with and without the three missing items during the study duration (N = 34). For each participant we determined the mean score of questionnaires with and without the three missing items and we calculated the difference between these mean values within all participants. A one-sample t-test showed that the mean difference between the mean questionnaire scores of questionnaires with and without missing items was not significantly different from zero (0.20, 95% CI [-0.24, 0.64], p = 0.367). Although the sample size for this additional analysis was relatively small, it's results appear to suggest that our method of replacing the missing items was valid. However, it remains possible that in response to falls the three missing items are filled out differently than the other FES-I items.

Furthermore, in our population we mostly recorded minor injuries after a fall (e.g. bruises and small cuts). This could be considered a limitation as falls resulting in more severe injuries may have a larger impact on concern about falling.

Finally, the lowest MMSE score of participants included in this study was 24 (N = 3). An MMSE score between 19 and 24 is generally considered an indication of "mild" cognitive impairment. It may be questioned whether these participants were able to fill out questionnaires and recall falls adequately. However, since there were only three participants with an MMSE score of 24 it is unlikely that this would have affected our conclusions.

Our findings imply that falls are not the primary cause of sustained concern about falling in older adults without a recent history of falling. Although concern about falling is increased up to one month after a fall, this effect was not seen after a period of up to a year. Clinicians and other healthcare professionals should be aware that there may be other factors besides recent falls, even those that result in injuries, that may cause older adults to become concerned about falling or that result in a sustained level of concern about falling.

## Conclusion

We observed only a small increase in concern about falling from the start to the end of a one-month interval in people who experienced a fall during this period and observed no increase from the baseline assessment to the end of the study one year later in people who experienced a fall during this period. From this, we conclude that older adults without a recent history of falling are quite resilient against developing concern about falling after having fallen. Furthermore, we found no evidence that high concern about falling increases the odds of falling in the following month nor year. We cannot advise concern about falling for use in fall prediction models, nor as a target in fall prevention programs aimed at older adults with little concern about falling who have no history of falling.

## Supporting information

**S1 Table. Relation between falling and concern about falling over long-term and short-term intervals.**
(PDF)

**S2 Table. Participant characteristics of both participants with and without a recent history of falling.**
(PDF)

**S3 Table. Relation between falling and concern about falling over long-term and short-term intervals in participants with a recent history of falling.**
(PDF)

**S4 Table. Relation between falling and concern about falling over long-term and short-term intervals in participants with and without a recent history of falling.**
(PDF)

## Acknowledgments

We thank all participants of the VIBE study as well as Mark Melman, Martine Rog and Babette Zwaard for their extensive help during data collection.

## Author Contributions

**Conceptualization:** Roel H. A. Weijer, Marco J. M. Hoozemans, Onno G. Meijer, Jaap H. van Dieën, Mirjam Pijnappels.

**Data curation:** Roel H. A. Weijer, Mirjam Pijnappels.

**Formal analysis:** Roel H. A. Weijer.

**Funding acquisition:** Mirjam Pijnappels.

**Investigation:** Roel H. A. Weijer.

**Methodology:** Roel H. A. Weijer, Marco J. M. Hoozemans, Jaap H. van Dieën, Mirjam Pijnappels.

**Project administration:** Roel H. A. Weijer, Mirjam Pijnappels.

**Resources:** Mirjam Pijnappels.

**Supervision:** Marco J. M. Hoozemans, Jaap H. van Dieën, Mirjam Pijnappels.

**Visualization:** Roel H. A. Weijer.

**Writing – original draft:** Roel H. A. Weijer.

**Writing – review & editing:** Marco J. M. Hoozemans, Onno G. Meijer, Jaap H. van Dieën, Mirjam Pijnappels.

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
