## [Decision Letter · Decision Letter 0]

27 Nov 2020

PONE-D-20-22235

The short and long term temporal relation between falls and concern about falling in older adults without a history of falling.

PLOS ONE

Dear Dr. Weijer,

Thank you for submitting your manuscript to PLOS ONE. After careful consideration, we feel that it has merit but does not fully meet PLOS ONE’s publication criteria as it currently stands. Therefore, we invite you to submit a revised version of the manuscript that addresses the points raised during the review process.

We had obtained three reviews, all of which indicate major issues to address before the manuscript could be further considered for publication. In particular, the points that were raised in relation to the methods are critically important. 

We look forward to receiving your revised manuscript.

Kind regards,

Catherine M. Capio

Academic Editor

PLOS ONE

Journal Requirements:

2.

We note that you have indicated that data from this study are available upon request. PLOS only allows data to be available upon request if there are legal or ethical restrictions on sharing data publicly. For more information on unacceptable data access restrictions, please see http://journals.plos.org/plosone/s/data-availability#loc-unacceptable-data-access-restrictions.

3. Please amend the manuscript submission data (via Edit Submission) to include author O.G. Meijer.

Reviewers' comments:

Reviewer's Responses to Questions

**Comments to the Author**

1. Is the manuscript technically sound, and do the data support the conclusions?

Reviewer #1: Yes

Reviewer #2: No

Reviewer #3: Yes

2. Has the statistical analysis been performed appropriately and rigorously? 

Reviewer #1: Yes

Reviewer #2: No

Reviewer #3: Yes

3. Have the authors made all data underlying the findings in their manuscript fully available?

Reviewer #1: No

Reviewer #2: No

Reviewer #3: Yes

4. Is the manuscript presented in an intelligible fashion and written in standard English?

Reviewer #1: Yes

Reviewer #2: Yes

Reviewer #3: Yes

5. Review Comments to the Author

Reviewer #1: This is a very interesting – and well-designed – study that explores both the short- and long-term relationship between concerns about falling (assed via the FES-I) and falls, both injurious and non-injurious. While previous research (e.g., Friedman et al.) has explored this relationship, there is need to isolate this exploration both temporally (ie, short- vs long-term) and in non-fallers. This research question has high levels of relevance for the clinical managing of individuals presenting concerns about falling. I commend the authors for conducting a clinically and scientifically important piece of research. I do, however, have a number of issues – some large, some small – which I feel need to be addressed before I can fully recommend the research for publication. Nonetheless, overall, this is a solid piece of work.

• Introduction – Line 49 – Should this not be “…we studied community-dwelling older adults without a RECENT history of falls”? As you only studied whether participants fell in the 12 months prior to your research, and not beyond this point. (So you may have had a participant who fell multiple times in the past few years, and simply did not fall in the 12 months prior to participation.) I would also recommend changing the abstract and title to reflect that the history of falling refers to a recent history of falling.

• On this point, I know that it is standard practice to assign participants to a faller/non-faller group based on the no. of falls over the past 12 months. But in my clinical experience, falls from beyond a 12-month period tend to also influence concern about falling and behavior that may either increase or decrease risk of falls – particularly if the fall was particularly serious/injurious. For example, an injurious fall 24 months ago may lead to a perpetual state of fear about falling; but due to experiencing a fall, the individual now attends physiotherapist-led exercise classes which improve balance and decrease the risk of future falls. Similarly, someone may have experienced a serious/injurious fall 2 years prior that drastically increased concerns about falling to the extent that any future falls leaves concerns unchanged (given the already high level of concerns). It might be worth adding this as a potential limitation.

• I feel that the Introduction is lacking in any description of how concern about concerns about falling may influence falls risk either indirectly (e.g., activity avoidance and deconditioning) or directly (e.g., adoption of ‘high-risk’ behaviors). For the indirect point, it would be worth referencing the paper by Hadjistavropoulos et al. 2011 (DOI: 10.1177/0898264310378039). For the direct point, it would be worth describing the work by Brown et al. (DOI: 10.1007/s00221-003-1468-7), Delbaere et al. (DOI: 10.1093/gerona/gln014) and Ellmers et al. (DOI: https://doi.org/10.1093/gerona/glz176). The reader needs to know how concerns of falling may influence fall risk.

• How was your sample size calculated?

• How valid is using minutes walking per day recorded at the very start of data collection (ie, before any falls) as a covariate for the relationship between variables throughout the study? Traditional models of fear of falling would suggest that following a fall, individuals will reduce their physical activity – meaning that any physical activity recording from pre-fall may no longer be valid. Is there a more appropriate physical functioning covariate that you could include instead? If possible, one that is likely to be more stable over time and less affected by the fall itself. Maybe functional balance (Berg balance), strength or even gait speed?

• The statistical analysis section is very well described, thank you for putting so much focus into this. The Results section were very clear as well. Excellent job.

• Table 1 – would it be possible to report FES values at the end of data collection? The baseline scores are very low (much lower than what we normally observe following clinical recommendation for falls prevention services). I am interested to see how these scores changed over the year.

• Do you have any other participant demographic data to report here? Berg scale? Timed up and go? Cognition? I also think it would be helpful to report the range for all variables.

• In previous work, Delbaere et al. (DOI: https://doi.org/10.1136/bmj.c4165) have shown that the relationship between concerns about falling and fall-risk may not be linear, and may instead be determined by physiological falls risk (which was not assessed or controlled in the present research). Can you please comment on how this may have affected your results?

• I would be very interested to see how the results differ for the participants with a recent history of falling. Would it be possible to conduct these analyses and attach them as a supplementary appendix? If these results show similar to the previous studies by Friedman et al. and Delbaere et al., then you have strong evidence for the reason for the discrepancy between your study and this previous work – it is due to the influence of fallers, in this previous work.

• In your reference list, reference number 6 is blank?

Reviewer #2: Overall

This research examined a cohort or elderly who had no falls within the past year, studying fear of falling and fall occurrence. However, a 1-year followup after falls was not completed (1-year data was estimated from linear regression) and statistical analysis methods were not supported in the text (analysis did not seem to relate to the research question). Also some missing data was manufactured instead of removing the trial. Based on these deficiencies, I stopped reviewing after methods since I cannot trust the results.

Abstract:

• I am not aware of a controversy for falls and fear of falling, best to remove this wording

Data sharing

• The statement about 15 year data storage is common for institutions since it indicates that long term data storage is required for studies and then the media can be destroyed. If your statement that this clause negates any data sharing then PLOS One is liable for all data sharing that has had institutional ethics review (i.e., likely every study with humans published). I expect that the authors are incorrect and that they should share the data as per PLOS One guidelines.

Introduction

• Line 64: what is meant by “the interrelations between falls and concern about falling”, what are interrelations in this context?

• Line 66: what is meant by “direction of relation”? is this a vector?

Methods

• How many people fell multiple times? How did you handle analysis for time period after multiple falls?

• You looked at 1 year total time, so some people may have fallen at the end of this period, how did you account for this? You did not really have 1 year followup after a fall (i.e. not long term after a fall). This must be fixed throughout the text and does affect interpretation in discussion.

• Line 117: By missing the last 3 questions, two of the activities were more likely to be rated of concern for people with a mobility issue (i.e., Walking up or down a slope, Walking on an uneven surface). Also the authors mentioned replacing 4 items so walking in crowds may also be missed. So, just averaging the other may underestimate the score. This should be considered and addressed in limitations.

• Age, gender, and activity level were used as covariants, but line 145 says missing covariate were replaced with the group mean value. This is inappropriate since you are essentially making up missing data that is specific to the participant. Participants without covariant data should be removed from the analysis.

• Line 151: replace “considered accounting for” with “accounted for” since “considered” implies that you thought about this but did not implement

• Line 162: It is unclear why regression is used and why predict? You have the data so no need to predict. As well, you indicate the interval is start to end of the year. So this does not involve falls directly, and you cannot day that this is long term 1 year response to falling.

• Line 176: Again, why regression?

• Line 193: The authors did not complete a 1 year followup, but tried to predict with an un-validated model. This method is inappropriate, real data is needed to answer the research question

Reviewer #3: I appreciate the opportunity to review this manuscript. This study examined the interrelationship between falling and concern about falling within a short term and a long term follow-up. Results showed that having falls or injurious falls was significantly associated with an increased concern about falling. Overall, the manuscript is well written. There are strengths and weaknesses in this study. My following comments include a summary of weaknesses and offer suggestions for the authors' consideration.

1. In the Abstract, the methods section should provide the type of statistical analysis. The results section should provide key findings (e.g., beta/odds ratio with 95% CI). The conclusion section should provide implication based on the study results.

2. The Introduction mainly focused on the study by Dalbaere et al about the relationship between falls and fear of falling. The authors should provide a bit more information on how fear of falling affects mobility and activities of daily living, and whether this might have been an underlying mechanism leading to future falls.

3. One of the inclusion criteria is MMSE score ≥ 19. What is the mean score (SD) of participants? This information should be included in Table 1. Meanwhile, a MMSE score between 19-24 is considered as “mild” cognitive impairment. Are participants with a score within this range able to recall fall history or fill in questionnaires? Please clarify.

4. Covariates in this study included age, gender, and the average walking duration. Were disease diagnoses, MMSE score, and use of mobility aid of participants also available? If so, they should be also included as covariates in the statistical models.

5. This study used an inertial sensor to measure daily activity (e.g., walking duration). How were these data associated with the FES-I scores? In addition, were participants assessed with sensors during the follow-up? It would be interesting to know if having fall(s) influences the physical activity level (e.g., walking duration) objectively measured by the inertial sensors.

6. In this study, nearly half of the participants experienced fall(s) over the one-year follow up. Is there any information about the types of falls? For example, was the fall caused by an internal factor (e.g., balance issue, postural hypotension) or an external factor (e.g., slippery floor, tripping hazard). This is important, as participants who fell due to an internal cause may have higher concern about falling compared to those who fell due to an external cause.

7. Among participants who fell, half of them experienced one fall and half experienced two or more falls. The analysis only examined the association between falling and concern about falling. The authors may want to investigate whether people having more falls would have a higher concern about falling than those having one fall.

8. In the discussion, the authors may want to highlight the importance of understanding the interrelationship of falling and concern about falling, and provide implications/ recommendations based on the study findings.

9. The authors stated that “a strength of the study is the exclusion of people with a self-reported history of falling”. However, the rate of falling in this study is quite high (nearly 50%) compared to the falling rate in community-dwelling older adults (about 30% in general). It is surprising to see that none of these participants had a fall history at baseline. This suggests the need to investigate what caused them to fall in this study (see comment #6). Please provide explanations.

6. PLOS authors have the option to publish the peer review history of their article (what does this mean?). If published, this will include your full peer review and any attached files.

Reviewer #1: No

Reviewer #2: No

Reviewer #3: No

---

## [Author Response · Author response to Decision Letter 0]

25 May 2021

Please see the included document "Response to Reviewers"

---

## [Decision Letter · Decision Letter 1]

4 Jun 2021

The short- and long-term temporal relation between falls and concern about falling in older adults without a recent history of falling.

PONE-D-20-22235R1

Dear Dr. Weijer,

We’re pleased to inform you that your manuscript has been judged scientifically suitable for publication and will be formally accepted for publication once it meets all outstanding technical requirements.

Kind regards,

Catherine M. Capio

Academic Editor

PLOS ONE

Additional Editor Comments (optional):

Reviewers' comments:

Reviewer's Responses to Questions

**Comments to the Author**

1. If the authors have adequately addressed your comments raised in a previous round of review and you feel that this manuscript is now acceptable for publication, you may indicate that here to bypass the “Comments to the Author” section, enter your conflict of interest statement in the “Confidential to Editor” section, and submit your "Accept" recommendation.

Reviewer #1: All comments have been addressed

Reviewer #3: All comments have been addressed

2. Is the manuscript technically sound, and do the data support the conclusions?

Reviewer #1: Yes

Reviewer #3: Yes

3. Has the statistical analysis been performed appropriately and rigorously? 

Reviewer #1: Yes

Reviewer #3: Yes

4. Have the authors made all data underlying the findings in their manuscript fully available?

Reviewer #1: Yes

Reviewer #3: Yes

5. Is the manuscript presented in an intelligible fashion and written in standard English?

Reviewer #1: Yes

Reviewer #3: Yes

6. Review Comments to the Author

Reviewer #1: I applaud the authors for conducting such a thorough revision (including new supplementary analyses), and for addressing my previous concerns. This paper adds to the current literature surrounding fear of falling and risk of falls, and I can now recommend it for publication.

Reviewer #3: The authors have addressed the reviewers' comments through major revision. The manuscript has been significantly improved. I have no further comments.

7. PLOS authors have the option to publish the peer review history of their article (what does this mean?). If published, this will include your full peer review and any attached files.

Reviewer #1: No

Reviewer #3: No

---

## [Editor Report · Acceptance letter]

1 Jul 2021

PONE-D-20-22235R1 

The short- and long-term temporal relation between falls and concern about falling in older adults without a recent history of falling. 

Dear Dr. Weijer:

I'm pleased to inform you that your manuscript has been deemed suitable for publication in PLOS ONE. Congratulations! Your manuscript is now with our production department. 

Kind regards, 

on behalf of

Dr. Catherine M. Capio 

Academic Editor

PLOS ONE